# Machine Learning Algorithms for Data Labeling: An Empirical Evaluation

## Abstract

The lack of labeled data is a major problem in both research and industrial settings since obtaining labels is often an expensive and time-consuming activity. In the past years, several machine learning algorithms were developed to assist and perform automated labeling in partially labeled datasets. While many of these algorithms are available in open-source packages, there is no research that investigates how these algorithms compare to each other in different types of datasets and with different percentages of available labels. To address this problem, this paper empirically evaluates and compares seven algorithms for automated labeling in terms of accuracy. We investigate how these algorithms perform in six different and well-known datasets with three different types of data, images, texts, and numerical values. We evaluate these algorithms under two different experimental conditions, with 10% and 50% labels of available labels in the dataset. Each algorithm, in each dataset for each experimental condition, is evaluated independently ten times with different random seeds. The results are analyzed and the algorithms are compared utilizing a Bayesian Bradley-Terry model. The results indicate that while the algorithms label spreading with K-nearest neighbors perform better in the aggregated results, the active learning algorithms query by instance QBC and query instance uncertainty sample perform better when there is only 10% of labels available. These results can help machine learning practitioners in choosing optimal machine learning algorithms to label their data.

## 1 INTRODUCTION

Supervised learning is the most commonly used machine learning paradigms. There are problems with supervised learning and machine learning in general. The first problem is that machine learning requires huge amounts of data. Secondly, supervised learning needs labels in the data. In a case study performed with industry, several labeling issues were found (Anonymous, 2020a).

A recent systematic literature review was conducted to see what type of machine learning algorithms exist to make the labeling easier. A recent systematic literature review investigated the use of Semi-supervised learning and Active learning for automatic labeling of data (Anonymous, 2020b). From those results the authors concluded which active and semi-supervised learning algorithms were the most popular and which datatypes they can be used on. However, even if there has been work done on active and semi-supervised learning, these learning paradigms are still very new for many companies and consequentially seldomly used.

Utilizing a simulation study we evaluated seven semi-supervised and active learning algorithms on six datasets of different types, numerical, text and image data. Implementing a Bayesian Bradley Terry model we ranked the algorithms according to accuracy and effort.

The contribution of this paper is to provide a taxonomy of automatic labeling algorithms and an empirical evaluation of algorithms in the taxonomy evaluated across two dimensions: *Performance*, how accurate the algorithm is, and *Effort*, how much manual work has to be done from the data scientist.

The remainder of this paper is organized as follows. In the upcoming section we provide the an overview about semi-supervised and active learning algorithms and how they work. In section 3 we will describe our study, how we preformed the simulations, what datasets and source code we used,

and what kind of metrics we used to evaluate performance, effort and applicability. In section 4 we provide the results from the simulation study and finally, we will interpret the results and conclude the paper in section 5.

## 2 BACKGROUND

### 2.1 ACTIVE LEARNING

Suppose a large unlabeled dataset is to be used for training a classification algorithm. *Active Learning (AL)*, poses query strategies on the data and selects points to be labeled according to a measure of informativeness called a *Query Strategy*. After the instances has been labeled with the help of the oracle, the machine learning algorithm is trained with this newly labeled data. If the learner thinks that the accuracy of the algorithm is too low and that the accuracy can be improved, the learner will request new and or replace some of the old labels. The algorithm will then be re-trained and evaluated once again. This procedure will continue iterative until some other stopping criteria has been reached. As a reference on AL, the reader is recommended to look at other sources such as (Settles, 2012). We shall now present the query strategies that we used in this text.

**Uncertainty Sampling** is according to (Anonymous, 2020b) the most commonly used active learning strategy. The idea of this approach is query the instances that we are the least certain about and then label these. Uncertainty sampling strategies are very commonly used and work especially well for probabilistic algorithms such as *logistic regression* according to (Lewis & Catlett, 1994). (Lewis & Catlett, 1994) concluded that uncertainty sampling has the ability to outperform random sampling by evaluating and comparing it to on a text classification dataset and (Joshi et al., 2009) concluded the same on image data by comparing accuracy scores on two uncertainty-sampling based methods and random sampling.

**Query-by-Committee**(QBC) means that we train a committee of classifiers and then query the instance on which the committee disagrees. Add the newly labeled instance to the labeled training data and retrain the algorithm on the new training set and repeat this procedure. What is important here is the way we measure disagreement. Some way to measure disagreement is through *entropy*, *vote-entropy* and *KL divergence*(Settles, 2012). QBC is relatively straightforward to implement and are applicable to any basic machine learning mode. (Seung et al., 1992) and (Freund et al., 1997) were the first to formulate QBC. In Seung et al. (1992) they use Monte Carlo simulation to show that QBC can outperform random sampling.

**Random sampling** is when the learner chooses to query the instances randomly and not according to any strategy. If a learner does not choose his query strategy carefully with respect to his data and machine learning algorithm, then active learning might not outperform choosing your instances randomly.

### 2.2 SEMI-SUPERVISED LEARNING

Semi-supervised machine learning is a class of machine learning algorithms that utilizes both labeled and unlabeled data. Semi-supervised algorithms are then trained on both the unlabeled and the labeled data and in some cases it even outperforms supervised classifiers. For more information on semi-supervised learning we refer the reader to (Zhu, 2005).
According to (Anonymous, 2020b) the second most popular semi-supervised learning algorithms are the *graph-based* algorithms. The idea of these algorithms is to build a graph from the training data. These graphs contains both labeled and unlabeled instances. Let each pair $(\mathbf{x}_i, y_i)$ and $(\mathbf{x}_j, y_j)$ represent each vertex and its corresponding label. Let the edge weight $w_{ij}$ represent the weight of the edge between vertex $i$ and vertex $j$. The larger $w_{ij}$ becomes the more similar are the labels of both vertices. The question is then how to compute the weight $w_{ij}$. Two examples of graph-based methods are *Label Propagation* and *Label Spreading* (Zha et al., 2009).

**Label propagation** was first introduced in (Zhu & Ghahramani, 2002) and presented as follows. Given labeled and unlabeled data, define the weight matrix $w_{ij}$. The probabilistic transition matrix $T$ is defined as the probability of jumping from vertex $i$ to vertex $j$

$$T_{ij} := P(j \rightarrow i) = \frac{w_{ij}}{\sum_{k=1}^{l+u} w_{kj}}.$$

The matrix $Y$ is called the label matrix and it's $i$th row represents the label probability distribution of vertex $x_i$ The label propagation algorithm consists of the following steps

1. All nodes propagate for one step: $Y \leftarrow TY$
2. Row normalize $Y$
3. Clamp the labeled data.

Repeat Step 1-2 until $Y$ converges. (Zhu & Ghahramani, 2002) evaluates the label propagation algorithm on both synthetic data and real-world classification data (Hull, 1994), by comparing it's error rates to that of $k$NN with $k = 1$. The results show that label propagation can outperform $k$NN when the number for labeled instances is greater than 40. Label propagation algorithms have used and evaluated in image annotation (Tang et al., 2011; Chua et al., 2009) and text classification (Pawar et al., 2016)

**Label Spreading** was first introduced in (Zhou et al., 2004). Given a partially labeled dataset with $c$ different labels. Let $\mathcal{F}$ be the set of all matrices of size $n \times c$ with non-negative entries. Let $F \in \mathcal{F}$. Each entry $F_{ij}$ in $F$ depends on how we label $x_i$. We have that

$$y_i = \arg\max_{j \leq c} F_{ij}$$

Define a matrix $Y \in \mathcal{F}$ such that

$$Y = \begin{cases} Y_{ij} = 1 \text{ if } y_i = j \\ Y_{ij} = 0 \text{ otherwise} \end{cases}$$

The label spreading algorithm is:

1. Define the affinity matrix

$$W = \begin{cases} W_{ij} = w_{ij} \text{ if } i \neq j \\ W_{ii} = 0 \end{cases}$$

2. Define the matrix $S = D^{-1/2}WD^{1/2}$ where $D$ is a diagonal matrix $D_{ii} = \sum_k W_{ik}$.
3. Iterate $F(t + 1) = \alpha SF(t) + (1 - \alpha)Y$ until convergence, $\alpha \in (0, 1)$.
4. Label $x_i$ as $y_i = \arg\max_{j \leq c} F_{ij}^*$ where $F^*$ is the limit of the sequence $\{F(t)\}$.

(Zhou et al., 2004) evaluates the label spreading algorithm on a toy dataset, images in the form of handwritten digits and text classification and concludes that it outperforms baseline models $k$NN with $k = 1$, SVM with RBF kernel.

## 2.3 THE BRADLEY TERRY MODEL

The Bradley-Terry model (Bradley & Terry, 1952; Cattelan, 2012) is one of the most commonly used models when it comes to analysis of paired comparison data between two objects $i$ and $j$ for $i, j = 1, ..., n$. The comparison can be by done several subjects $s = 1, ..., S$ and the total number of possible paired comparisons is equal to $n(n - 1)/2$. Let $y_s = (y_{s,1,2}, ..., y_{s,n-1,1})$ be the vector of outcomes of all paired comparisons, we will assume that we outcomes are independent.

Let $\mu_i \in \mathbb{R}, i = 1, 2, ..., n$ denote a latent "strength" of the algorithm being compared. If the paired comparison can have only two outcomes and ties are randomly resolved, the probability of $i$ beating $j$ can be represented by:

$$P[i \text{ beats } j] = \frac{e^{\mu_i}}{e^{\mu_i} + e^{\mu_j}}$$

Reducing the expression to a logistic regression (Bradley & Terry, 1952):

$$P(i \text{ over } j) = \text{logit}^{-1}(\mu_i - \mu_j)$$

By estimating the strength latent variable $\mu$, we can infer the probability of one algorithm to beat the other and use this information to rank the algorithms

## 3 RESEARCH METHOD

In this section we present the details about the datasets that we used for our simulations, the experimental conditions and the used algorithms.

The goal of this study is to show in detail how machine learning algorithms can be used to help with data labeling and to provide an in-depth comparison on how these different algorithms perform on different types of data. To achieve this we performed an empirical evaluation of seven different active learning and semi-supervised learning algorithms and evaluated them on six datasets under different conditions.

The main research questions that we use to evaluate the machine learning algorithms are the following.

- **RQ1:** How can we rank different active learning and semi-supervised learning algorithms in terms of accuracy?
- **RQ2:** How do the rank of these algorithms with changes in the amount of manual labeling effort prior to applying these methods?

### 3.1 SIMULATIONS

As recognized in (Anonymous, 2020b) Co-training/multi-view learning are the most popular algorithms but are based on the assumption than we can watch an instance from multiple views. Graph-based algorithms are the second most common type of semi-supervised learning algorithm. Uncertainty sampling methods are very popular active learning query strategies followed by QBC.

Furthermore we have included two different graph-based algorithms Label Spreading and Label Propagation. Both methods are easy to implement using python an

- **Label Spreading using $k$-NN** is implemented with $w_{ij} = kNN$, $k = 7$, $\alpha = 0.2$ (Pyt, b).
- **Label Spreading using RBF** is implemented with $w_{ij} = \exp(-\gamma|x_i - x_j|^2)$, $\gamma = 20$, $\alpha = 0.2$
- **Label Propagation using $k$-NN** is implemented with $w_{ij} = kNN$, $k = 7$ (Pyt, a).
- **Label Propagation using RBF** is implemented with $w_{ij} = \exp(-\gamma|x_i - x_j|^2)$, $\gamma = 20$
- **Radnom Sampling, Uncertainty Sampling and QBC:** Each dataset was randomly split into training and test set, unlabeled and labeled set. 80% of the data was allocated for training and 20% was allocated for testing. As a stopping criterion we choose to stop after 50 instances had been queried.

We choose six benchmarked datasets to be used in our experiments. Two numerical datasets, two text datasets and two image datasets. Due to the size of some datasets and to limited time and computational resources required we had to reduce the number of images used in our experiments. However, we made sure we used the same ration for the classes to get a fair estimated.

- **Image data:**
  - **Cifar-10:** This dataset originally contains 60000 32x32 colored images that can be divided into ten classes, airplane, automobile, bird, car, deer, dog, frog, horse, ship and truck (cif).
  - **Digits:** This dataset contains 1797 samples of 8x8 images containing one digit each. There are ten classes that represent which digits is contained in each image. (dig)
- **Text data:**
  - **Fake and true news:** This is a dataset containing 44594 instances and 5 features. The features are, "title", the title of the news article. "text", the text of the article, "subject" the article subject and a column representing the label classes, "False" or "Truthful". From this dataset we only extracted the "text" column and used it as a features to predict the labels. The dataset can be download from Kaggle (fak).

  - **20news:** This dataset contains 18846 instances divided into 20 classes that describes the 20 different types of news. (20n).
- **Numerical data**
  - **Iris:** This dataset is a classic example for multi-class classification. It contains 150 instances across three classes.(iri).
  - **Wine:** The wine dataset also a classic example of multi-class classification. It contains 178 instances across three classes.(win).

For each dataset we ran each iteration ten times with different random seeds. Furthermore, the only parameter that we change is number of labeled instances. To answer RQ2 we have to vary the amount of instances in dataset that are already labeled. In our experiments we choose 10% to represent small amount of manual effort required and 50% for large amount of effort required. From each iteration we logged the $F_1$-score to measure the accuracy of our predictive labels.

## 4 RESULTS

From the simulations a dataset of 840 instances was collected. To analyze this data, we first rank each algorithm in by each of the ten iterations of each dataset in each experimental condition. This data is them expanded into paired comparisons for the use in the Bradley-Terry model (Turner et al., 2020; Cattelan, 2012). In this model, $y$ is a binary variable that indicates which algorithm beats the other:

$$y \sim \text{Bernoulli}(p),$$
$$p = \text{logit}^{-1}(\mu_{\text{algo1}} - \mu_{\text{algo0}}),$$
$$\mu_i \sim \text{Normal}(0, 5).$$

The same model is used to analyze both research questions. The model is written in Stan (Carpenter et al., 2017), which implements the No U-turn Hamiltonian Monte Carlo sampler (Hoffman & Gelman, 2014). We utilize the following configurations: 4 chains, warm-up of 200 iterations and a total of 2000 iterations. The data transformation, tables, plots and assessing the convergence of the chains are conducted in R together with package `rstan` and the collection of packages `tidyverse`.

The prior distributions of the $\mu_i$ parameters are adjusted to be weakly-informative distributions. The presented model estimates the posterior distribution of the latent strength parameters $\mu_i$. In turn, sampling and ranking over the posterior distribution of the strength parameters allows us to obtain a posterior distribution of the ranks.

### 4.1 AGGREGATED RESULTS

The descriptive statistics of these seven datasets are summarized in Table 1. Table 1 contains the mean, standard deviation, median as well as 5% and 95% quantiles for each method. Figure 1 provides descriptive statistics in the form of a boxplot.

Table 1: Summary statistics for the accuracy of the aggregated data. 5% and 95% represents the quantiles.

| Model | Mean | SD | Median | 5% | 95% |
|---|---|---|---|---|---|
| LabelPropagationKNN | 0.328 | 0.288 | 0.209 | 0.018 | 0.967 |
| LabelPropagationRBF | 0.367 | 0.370 | 0.165 | 0.018 | 0.987 |
| LabelSpreadingKNN | 0.750 | 0.286 | 0.829 | 0.099 | 0.987 |
| LabelSpreadingRBF | 0.578 | 0.316 | 0.595 | 0.128 | 0.993 |
| QBC | 0.775 | 0.281 | 0.910 | 0.150 | 0.974 |
| RandomSampling | 0.758 | 0.277 | 0.899 | 0.146 | 0.973 |
| UncertaintySampling | 0.783 | 0.284 | 0.923 | 0.157 | 1.000 |

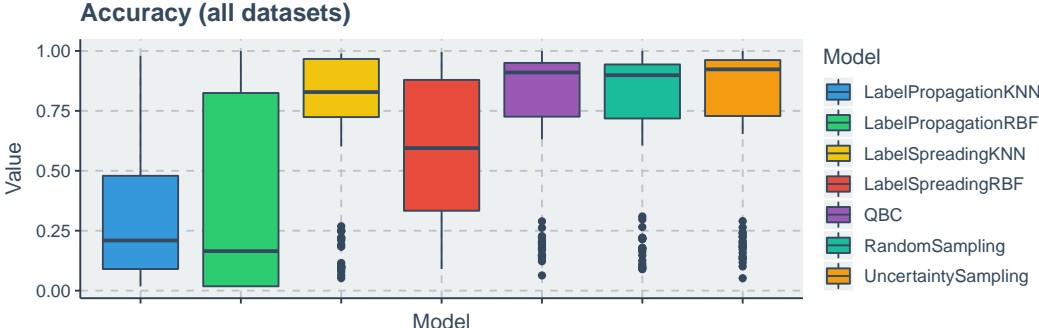

Figure 1: Boxplot of all algorithms

Based on the Bradley-Terry model described above, the parameter strength is computed for each algorithm. Figure 2 illustrates the distribution of the parameter strengths along with their High Posterior Density interval. To rank the algorithms we sample over the posterior distribution of the strength parameters 1000 times. The median ranks and their corresponding variances is displayed in Table 3.

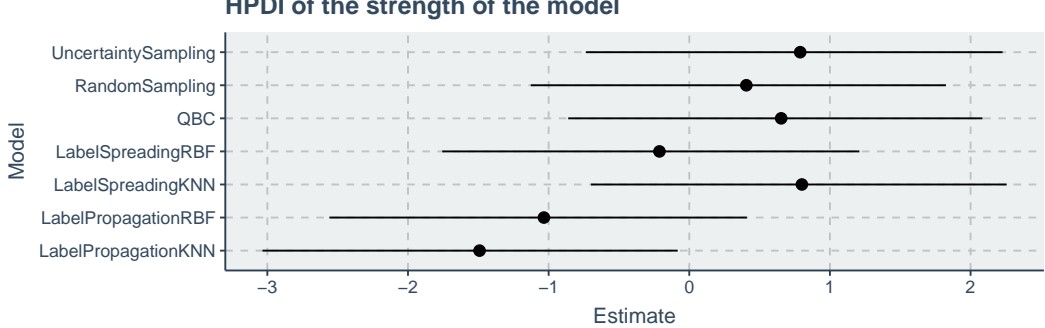

Figure 2

Table 2: Ranking of the algorithms

| Models | Median Rank | Variance of the Rank |
|---|---|---|
| LabelSpreadingKNN | 1 | 0.377 |
| UncertaintySampling | 2 | 0.416 |
| QBC | 3 | 0.229 |
| RandomSampling | 4 | 0.006 |
| LabelSpreadingRBF | 5 | 0.000 |
| LabelPropagationRBF | 6 | 0.000 |
| LabelPropagationKNN | 7 | 0.000 |

## 4.2 MANUAL EFFORT

The descriptive statistics of the seven datasets are located in Table 3. Table 3 contains the mean, standard deviation, median as well as 5% and 95% quantiles for each method. Figure 3 provides descriptive statistics in the form of two boxplots, one for 10% and one for 50% labels.

Based on the Bradley-Terry model described above, the parameter strength is computed for each algorithm. Figure 4a and Figure 4b illustrate the distribution of the parameter strengths along with their High Posterior Density interval for 10% and 50% labels respectively. To rank the algorithms we sample over the posterior distribution of the strength parameters 1000 times. The median ranks

### Accuracy (all datasets)

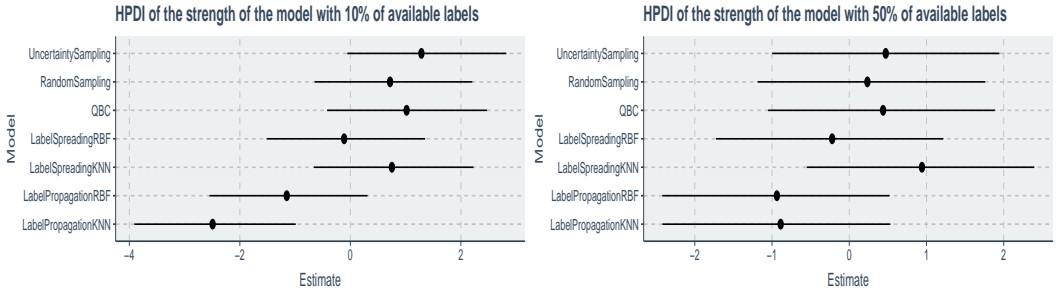

Figure 3: Boxplots of algorithms, The left boxplot is for 10% available lables, the right boxplot is for 50% available labels.

Table 3: Summary statistics for the accuracy aggregated data

| Model | Mean | SD | Median | 5% | 95% |
|---|---|---|---|---|---|
| **10% labels available** | | | | | |
| LabelPropagationKNN | 0.135 | 0.111 | 0.128 | 0.018 | 0.333 |
| LabelPropagationRBF | 0.301 | 0.332 | 0.165 | 0.018 | 0.961 |
| LabelSpreadingKNN | 0.718 | 0.304 | 0.796 | 0.090 | 0.981 |
| LabelSpreadingRBF | 0.518 | 0.316 | 0.444 | 0.128 | 0.982 |
| QBC | 0.768 | 0.280 | 0.900 | 0.174 | 0.974 |
| RandomSampling | 0.740 | 0.272 | 0.864 | 0.161 | 0.967 |
| UncertaintySampling | 0.781 | 0.278 | 0.917 | 0.170 | 1.000 |
| **50% labels available** | | | | | |
| LabelPropagationKNN | 0.521 | 0.281 | 0.482 | 0.136 | 0.973 |
| LabelPropagationRBF | 0.433 | 0.396 | 0.346 | 0.018 | 0.987 |
| LabelSpreadingKNN | 0.782 | 0.265 | 0.873 | 0.218 | 0.988 |
| LabelSpreadingRBF | 0.638 | 0.306 | 0.706 | 0.132 | 0.994 |
| QBC | 0.782 | 0.284 | 0.918 | 0.146 | 0.974 |
| RandomSampling | 0.777 | 0.282 | 0.902 | 0.146 | 1.000 |
| UncertaintySampling | 0.784 | 0.292 | 0.933 | 0.156 | 1.000 |

and their corresponding variances is displayed in Table 4 and Table 5 for 10% and 50% labels respectively.

(a) The HPDI interval of the estimated strength parameters of the algorithms with 10% available labels

(b) The HPDI interval of the estimated strength parameters of the algorithms with 50% available labels

Table 4: Ranking of the algorithms with 10% of available labels

| Models | Median Rank | Variance of the Rank |
|---|---|---|
| UncertaintySampling | 1 | 0.051 |
| QBC | 2 | 0.157 |
| LabelSpreadingKNN | 3 | 0.324 |
| RandomSampling | 4 | 0.299 |
| LabelSpreadingRBF | 5 | 0.000 |
| LabelPropagationRBF | 6 | 0.000 |
| LabelPropagationKNN | 7 | 0.000 |

Table 5: Ranking of the algorithms with 50% of available labels

| Models | Median Rank | Variance of the Rank |
|---|---|---|
| LabelSpreadingKNN | 1 | 0.004 |
| UncertaintySampling | 2 | 0.353 |
| QBC | 3 | 0.359 |
| RandomSampling | 4 | 0.161 |
| LabelSpreadingRBF | 5 | 0.003 |
| LabelPropagationKNN | 6 | 0.234 |
| LabelPropagationRBF | 7 | 0.234 |

## 5 CONCLUSION

According to Table 2, Label Spreading using $k$NN is the highest ranking algorithm followed by uncertainty sampling, QBC and then random sampling. The uncertainty intervals of the posterior distribution are shown in Figure 2. The large overlap between the top three algorithms strength parameters indicates the uncertainty in rank between them (which can be observed in the large variance of each rank).

According to Table 4, the highest ranking algorithm when having access to 10% available labels is uncertainty sampling, followed by QBC, label spreading using $k$NN and random sampling. When having access to 50% labels the highest ranking algorithm is label spreading using $k$NN, followed by uncertainty sampling, QBC, and random sampling according to Table 5. The uncertainty intervals of the posterior distribution are shown in Figures 4a and 4b for 10% and 50% respectively. The overlap between the top algorithms strength parameters indicates the uncertainty in their estimates, this can also be observed in their variance.

The goal of this study is to provide a detailed overview of what machine learning algorithm should be used for automatic labeling of data in industrial contexts. Based on the results, the top four algorithms are label spreading using $k$NN, uncertainty sampling, QBC and random sampling. For the aggregated results as well as when having access to 50% labeled data, the highest ranking algorithm is label spreading using $k$NN. However, when 10% labels are available, Uncertainty Sampling ranks highest followed by QBC. Thus this paper contributes in assisting machine learning practitioners to choose the optimal machine learning algorithm for automatic labeling. In future work, simulations will include more datasets to provide a better understanding of how well algorithms perform on different types of data.

## ACKNOWLEDGMENT

This work was partially supported by the Wallenberg AI Autonomous Systems and Software Program (WASP) funded by Knut and Alice Wallenberg Fundation.

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
