# OpenReview forum: "Machine Learning Algorithms for Data Labeling: An Empirical Evaluation"
_ICLR.cc/2021/Conference — Reject_

### Official Review · AnonReviewer2 · 2020-10-22
**The paper compares several approaches for labelling data, spanning active learning and semi-supervised strategies. The problem and choice of algorithms are not sufficiently motivated. In particular, it is unclear why the authors compare semi-supervised techniques with active learning techniques. The latter requires an oracle while the former doesn't. There is a range of further largely unjustified ad-hoc choices. This work is not sufficiently significant to warrant publication.**

**Rating:** 3
**Confidence:** 4

**Review:**



## Detailed Comments
- "There are problems with supervised learning and machine learning in general." - This statement is so general, it is essentially vacuous.
- "machine learning requires huge amounts of data" - unclear what this means. and whatever it means it's not true in general.
- "severe labeling issues were found" - what issues?
- "we provide the an overview"
- "(AL)," - missing space. this happens at several places in the text
- "continue iterative" - iteratively
- "other stopping criteria" -> "criterion". also, why "other"? there was no stopping criterion mentioned so far.
- "If a learner does not choose his strategy" -> their strategy
- eq at bottom of page 2: what is "u"?

---

### Official Review · AnonReviewer3 · 2020-10-27
**An important topic (empirical evaluation) but it might require a major revision**

**Rating:** 4
**Confidence:** 4

**Review:**

In this paper, the authors present an empirical analysis of seven machine learning algorithms based on six benchmark datasets: Four graph-based semi-supervised learning algorithms and three active learning algorithms have been evaluated on image data sets (CIFAR10, Digits), texts (Fake and true news, 20news), and other data types (Iris and Wine). Based on the empirical performances of these algorithms, the authors provided a ranked list of the studied algorithms.

Empirical comparisons of multiple machine learning algorithms on real-world datasets are important as they help understand the strengths and weaknesses of different algorithms when they are deployed in real-world environments. However, I think this paper will benefit significantly from a major revision addressing the concerns listed below:
1) Algorithm choices: all algorithms considered in the current paper were already extensively studied. Focusing on state of the art approaches (including recent deep learning-based approach) could significantly strengthen the practical relevance and impact of experiments and conclusions reported here.
2) Data set choices: more challenging datasets can be considered: The datasets used in the current paper (including classical Iris and Wine) have up to only 20 categories.
3) Presentation: I found it challenging to comprehend the exact experimental settings.
- Section 3.1 states that 80% of data are allocated for training while the remaining 20% are used in testing. How does this setting apply for active learning algorithms? Did they actively select data points to label until when 80% are labeled or use only `50 instances’?
- How is 80/20% decomposition applied to semi-supervised learning algorithms? Did they use the entire dataset with 80% of the entities labeled? In typical application scenarios of semi-supervised learning, only small portions of data instances are labeled. I was not sure if the experimental setting prepared in the current paper reflects well the real-world application scenarios of semi-supervised learning.
- Also, in general, semi-supervised learning and active learning are different problems. I was not sure how the results reported in the current paper including Table 1 should be interpreted: I guess this does not suggest that semi-supervised learning algorithms are better than active learning approaches.
4) Without having access to two `anonymous’ papers cited in this submission, it is hard to properly assess the contributions of this paper.

---

### Official Review · AnonReviewer1 · 2020-10-27
**Good idea, but flawed execution**

**Rating:** 4
**Confidence:** 4

**Review:**

The paper presents an empirical comparison of different approaches for data
labeling. The authors describe their experimental setup and findings, making
recommendations for when to use what approach in practice.

The authors reference their own anonymous work throughout the paper as
justification for the presented investigation and its parameters. This is
problematic as the reviewers are now unable to confirm that the presented
investigation is well-grounded.

The authors evaluate their approaches on only six datasets. It is unclear to
what extent the results generalize, in particular as no detailed results per
dataset are given. There could be significant differences between the different
types of datasets, but not enough data is presented to judge. This matters in
particular with respect to the recommendations the authors make at the end of
the paper.

Some details of the experimental setup are unclear. The authors say that they
measure F1 score, but then refer to accuracy (e.g. in Figure 1). Which measure
was used? The experimental setup describes six datasets, but the results text
refers to seven. The results presented in Table 1 and Figure 1 seem to disagree
with Table 2 -- LabelSpreadingKNN is the highest-ranked algorithm, but
UncertaintySampling performs better in terms of all the statistics presented in
Table 1. The same is true for the second set of experiments (Tables 3 and 4).
For the first set of experiments it is unclear what fraction of labels were
missing.

It is unclear why the Bradley-Terry model was used here to compare outcomes.
There are multiple other methods to judge how and whether paired distributions
differ. It appears that only ranks were used for this comparison and not the
actual performance numbers.

Finally, all methods evaluated by the authors have hyperparameters that need to
be set. It is unclear how the authors chose the particular values they used in
the experiments, and tuning them for best performance may have a major impact on
their performance and the rankings. Conclusions from untuned methods are
unlikely to generalize.

There are numerous typos and grammatical mistakes throughout the paper.

---

### Official Review · AnonReviewer4 · 2020-10-28
**A weak empirical evaluation**

**Rating:** 3
**Confidence:** 5

**Review:**

This paper aims to evaluate the performance of seven automated labeling algorithms in terms of accuracy. The authors conducted a set of experiments on six datasets from different domains under two typical settings where 10% and 50%of labels in the datasets are available. Experimental results show that the algorithms label spreading with KNN perform better in the aggregated results, the active learning algorithms  QBC and query instance uncertainty sample perform better when 10% of labels available.

Overall, this paper cannot meet the high-quality requirements of ICLR.  First, active learning algorithms such as QBC and uncertainty sampling is not automated labeling algorithms. They are only strategies for the selection of unlabeled instances. The selected instance either can be labeled by human experts or automated labeling algorithms. Second, when evaluating an automated labeling algorithm, merely using accuracy is not enough. For example, when the underlying class distributions are imbalanced, the accuracy is not sufficient to characterize the generalization performance of a learning algorithm. Third, two settings of 10% and 50% of labels available are also insufficient. Many papers of the empirical study investigated the performance under more complicated settings. Finally, the number of investigated methods is two small and the paper should cover more state-of-the-art algorithms.

---

### Decision · Program_Chairs · 2021-01-07
**Final Decision**

**Decision:**

Reject

**Comment:**

The paper evaluates several different strategies for labeling of missing data, and recommend the best strategy in practice based on the empirical results on six data sets.

The reviewers agree that empirical evaluation is important for providing a good guideline for this practical problem. The concerns of the reviewers include the lack of motivation on the chosen strategies, the lack of novelty (the tools in the strategies are all pretty standard in the literature), the lack of reproducibility (by referring to the authors' own anonymous work for parameters), and the lack of breadth (e.g #data sets) and depth (e.g. metrics explored) in the experiments.